# Investigation of Functionalized Surface Charges of Thermoplastic Starch/Zinc Oxide Nanocomposite Films Using Polyaniline: The Potential of Improved Antibacterial Properties

**DOI:** 10.3390/polym13030425

**Published:** 2021-01-28

**Authors:** Korakot Charoensri, Chatchai Rodwihok, Duangmanee Wongratanaphisan, Jung A. Ko, Jin Suk Chung, Hyun Jin Park

**Affiliations:** 1Department of Biotechnology, College of Life Sciences and Biotechnology, Korea University, 145 Anam-ro, Seongbuk-gu, Seoul 02841, Korea; korapop253@korea.ac.kr (K.C.); elfinja@korea.ac.kr (J.A.K.); 2School of Chemical Engineering, University of Ulsan, 93 Daehak-ro, Nam-gu, Ulsan 680-749, Korea; c.rodwihok@hotmail.com; 3Department of Physics and Materials Science, Faculty of Science, Chiang Mai University, Chiang Mai 50200, Thailand; duangmanee.wong@cmu.ac.th

**Keywords:** thermoplastic starch, Functionalized ZnO NPs, Polyaniline, positive surface charge, antibacterial activity

## Abstract

Improving the antibacterial activity of biodegradable materials is crucial for combatting widespread drug-resistant bacteria and plastic pollutants. In this work, we studied polyaniline (PANI)-functionalized zinc oxide nanoparticles (ZnO NPs) to improve surface charges. A PANI-functionalized ZnO NP surface was prepared using a simple impregnation technique. The PANI functionalization of ZnO successfully increased the positive surface charge of the ZnO NPs. In addition, PANI-functionalized ZnO improved mechanical properties and thermal stability. Besides those properties, the water permeability of the bionanocomposite films was decreased due to their increased hydrophobicity. PANI-functionalized ZnO NPs were applied to thermoplastic starch (TPS) films for physical properties and antibacterial studies using Escherichia coli (*E. coli*) and Staphylococcus aureus (*S. aureus*). The PANI-functionalized ZnO bionanocomposite films exhibited excellent antibacterial activity for both *E. coli* (76%) and *S. aureus* (72%). This result suggests that PANI-functionalized ZnO NPs can improve the antibacterial activity of TPS-based bionanocomposite films.

## 1. Introduction

Recently, there has been growing concern regarding the threat of microorganism contamination. Some bacteria have developed resistance to widely used antibacterial agents [1,2]. Additionally, environmental problems related to plastic pollution are consequential issues that are mainly attributed to the use of single-use plastic products [3,4]. To combat bacterial issues, many researchers have attempted to improve the efficiency of antimicrobial agents in various applications. The main function of packaging is to protect products from contamination. Therefore, the improvement of antibacterial agents for packaging applications is an attractive topic [5,6]. Many researchers have focused on the development of metal/metal oxide nanocomposite-assisted packaging polymers (e.g., silver nanoparticles (Ag NPs), silver oxide (AgO), copper oxide (CuO), and zinc oxide (ZnO)) for active packaging applications to provide enhanced physicochemical and antibacterial properties [7,8].

ZnO NPs are promising materials with high chemical and thermal stability [9,10] as well as low toxicity which are generally recognized as safe materials by the US Food and Drug Administration [11]. ZnO NPs also provide a strong ultraviolet (UV) barrier [12,13] and excellent compatibility in many applications, including drug delivery, tissue engineering, immunotherapy, gene delivery, biosensing, and composite films [11,14,15,16,17]. The incorporation of ZnO NPs has been reported to have a positive effect on the antibacterial activity of biopolymers, including soy protein isolated, chitosan, and starch [4,7,18,19]. Based on recent literature review, many researchers have studied the improvement of ZnO NPs antibacterial activity by modifying their surface charges. Doping techniques using metal/metal oxide NPs and charged organic compounds have been applied to modify ZnO NPs’ surface charges while maintaining particle size and colloid activity [20]. Anionic molecules such as amino acids and citric acid have been reported to improve the surface charge tunability of ZnO NPs [21,22]. Cationic molecules such as dido-decyl dimethyl ammonium bromide, L-serine, and cerium have been reported to preserve or increase the positive surface charge of ZnO NPs [20,23,24,25]. Additionally, the antibacterial activity of gram-negative and gram-positive bacteria has been leveraged in studies on modifying the surface charges of ZnO NPs. Regarding the sanitation and contamination of microorganisms, ZnO has excellent potential for active packaging and medical coating applications based on its antibacterial activity, which is attributed to photocatalytic activity and reactive oxygen species (ROS) production [26,27].

Thermoplastic starch (TPS) is one of the most promising natural polymer materials for packaging applications based on its inherent biodegradability, abundance, and affordability [28,29,30]. TPS consists of amylose and amylopectin, and the ratio of these components has a significant impact on film polymerization, and chemical and physical properties [31,32]. As a natural polymer material, corn starch (CS) has 28 wt% of amylose content, which is greater than that of cassava starch (18.6 wt%) and tapioca starch (16.7 wt%) [33]. Therefore, CS has been recognized as a good candidate in biodegradable plastic and single-use products, such as packaging for syringes, needles, wrapping, cutlery, straws, bags. However, biodegradable starch films have some drawbacks in terms of hydrophilicity, mechanical properties, and processability [34,35,36]. To the best of our knowledge, there have been no reports on the antimicrobial activity of polyaniline (PANI)-functionalized ZnO NP bionanocomposite films.

In this study, we focused on the preparation and characterization of PANI-functionalized ZnO bionanocomposite film for antibacterial applications. The amount of PANI additive was varied to study its effects of improved surface properties and antibacterial activities of the prepared bionanocomposite films which could potentially be used as an alternative polymer material for packaging application.

## 2. Materials and Methods

### 2.1. Materials

Zinc acetate dihydrate (Zn (CH_3_COO)_2_·H_2_O) was purchased from samchun pure chemical (Gyeonggi, South Korea), Sodium hydroxide (NaOH) was purchased from Daejung chemicals metals (Gyeonggi, South Korea). Aniline was from sigma aldrich (Gyeonggi, South Korea). Extra pure grade starch corn, glycerin and glacial acetic acid were purchased from Duksan Scince (Gyeonggi, South Korea). Mueller Hinton agar, LB broth (Luria-Bertani), Tryptic soy agar were obtained from Difco (Sparks, MD, USA). Deionized (DI) water was used to prepare and clean samples. All chemicals were utilized as received in a simple synthesis process without any further purification.

### 2.2. ZnO and Functionalized ZnO NP Synthesis and Characterization

ZnO NPs were prepared as described in a previous study [24,37] using hydrothermal techniques. In summary, 0.4 M of Zn (CH_3_COO)_2_·H_2_O was dispersed in 170 mL of DI water under continuous magnetic stirring. NaOH solution was prepared separately in DI water and added to the prepared zinc solution to adjust its pH to 10. After achieving the desired pH, this mixture was transferred into an autoclave and heated to 180 °C for 12 h. The precipitate was collected via filtration and rinsed several times with DI water. The sample was then dried at 70 °C under vacuum for 24 h.

PANI solutions were prepared by mixing precalculated amounts of PANI (1, 2, and 3 mol/L) into an ethanol/DI (4:1) solution under stirring for 30 min. The resulting solutions are referred to ZP1, ZP2, and ZP3, respectively. To functionalize PANI onto ZnO samples, 0.5 g of a ZnO sample was immersed in a PANI solution and stirred for 12 h. Finally, as-prepared powders were collected and dried using the same procedure as that used for the original ZnO samples.

The ZnO and functionalized ZnO NPs were chemically characterized using Fourier-transform infrared spectroscopy (FT-IR, Thermo Electron Co., Waltham, MA, USA). X-ray photoelectron spectroscopy (XPS, K-alpha; Thermo Fisher Scientific Co., Waltham, MA, USA) was utilized to investigate the chemical interactions and valence states of the samples. X-ray diffraction (XRD, Rigaku D/MAZX 2500V/PC model, Tokyo, Japan) was used to investigate the phase structures of the samples. Field-emission scanning electron microscopy (FE-SEM, JEOL-JSM-7600F, Tokyo, Japan) with Energy Dispersive X-Ray Spectroscopy and and high-resolution transmission electron microscopy (HR-TEM, JEM-2100F-JEOL, Japan) with exwas used to observe the surface morphologies of the samples. Average particle size and surface charge (zeta potential, Zetasizer Nano ZS, MALVERN, Gyeonggi, South Korea) were investigated using a zeta potential analyzer.

### 2.3. Preparation of Bionanocomposite Film and Characterization

The solution casting method was utilized to prepare neat CS films and bionanocomposite films (referred to as CS, CS/ZP0, CS/ZP1, CS/ZP2, and CS/ZP3). Initially, aqueous solutions containing 3% CS were obtained by mixing 3 g of CS into 100 mL of DI water. Next, 100 mg of as prepared ZnO and functionalized ZnO powders were added to the solutions with 1 g of glycerol as a plasticizing agent. We adjusted the pH values of the solutions to achieve final pH values of 3–4 using 5% acetic acid solution. Next, the solutions were gradually heated under continuous stirring at 300 rpm for 30 min to facilitate full gelatinization of the CS. After heating, the mixtures were degassed in a vacuum oven. The mixtures were then cooled and poured evenly over Petri dishes and evaporated at 25 ± 1 °C for 48 h. All dried sample films were stored in a thermos-hygrostat chamber (25 °C, RH = 50%) for further analysis.

Chemical functional group identification of the sample films was performed using FTIR in attenuated reflectance mode (Cary 630; Agilent Technologies, Inc., Santa Clara, CA, USA). The surface morphologies of the bionanocomposite films were analyzed using a scanning electron microscope (FE-SEM, JEOL-JSM-7600F, Tokyo, Japan). The crystallinity of the samples was evaluated using an X-ray diffractometer (XRD, Rigaku D/MAZX 2500V/PC model, Tokyo, Japan).

### 2.4. Functional Properties of Bionanocomposite Film Analysis

Film thickness was determined using a micrometer (Mitutoyo Co. Ltd., Tokyo, Japan) with 0.001 mm accuracy at five random positions around the samples. The solubility was interpreted as the percentage of solubilized dry basis after dispersion in DI water at 25 °C for 24 h. The square samples (20 mm × 20 mm) from each sample were cut and weight. The amount of dry basis in the initial and final samples was measured by drying the samples at 105 °C for 24 h. The solubility in water was calculated as follows:(1)$$\%\;Solubility=\frac{W_{0}-W_{1}}{W_{0}}\times 100$$
where *W*_0_ is the weight of the initial dried samples, and *W*_1_ is the weight of the dried sample after immersion. The water vapor permeability (*WVP*) was determined using a cup method following the ASTM E96-80 standard. The film sample was mounted on a cup that contains DI water (21 mm in depth and 46 mm in diameter) and using an O-ring to cover and prevent leakage. The assembled cups were always stored in a chamber set at 25 °C and 50% RH and weighed at 6 h intervals for 48 h to measure the water vapor transmission rate (*WVTR*). The *WVP* was calculated as follows:(2)$$WVP=\frac{WVTR\times d}{\Delta P}$$
where *d* (mm) is the average of the sample film thickness, and Δ*P* (Pa) is the partial water vapor pressure of the two sides across the film. The average value was calculated by repeating the experiment three times. The tensile strength (TS) and elongation at break parameters of nanocomposite film samples were determined using a universal testing machine (Model 3366, Instron Engineering Co., Norwood, MA, USA) operated with a load cell of 5 kN, an initial grip separation of 50 mm, and a cross-head speed of 500 mm/min, according to the ASTM 0882-02 standard test method. The degradation process of the nanocomposite film was conducted by a thermogravimetric analyzer (TGA, N-1000, SCINCO, Seoul, Korea) using a nitrogen atmosphere with a flow rate of 50 mL/min, and the sample was heated from 30 to 500 °C at a rate of 20 °C/min. The weight loss as a function of temperature was analyzed.

### 2.5. Antibacterial Activity

An agar disc diffusion assay was utilized to screen samples that provided relatively strong antibacterial activity against *Escherichia coli* (*E. coli*) and *Staphylococcus aureus* (*S. aureus*). Five mixtures of bacteria were incubated at 37 °C overnight and diluted in Luria-Bertani (LB) broth to achieve a population of 5–6 log CFU/mL. Film samples (6 mm in diameter) were placed on Mueller–Hinton agar (MHA) plates covered with 0.1 mL of each type of bacteria. Next, the plates were incubated at 37 °C for 24 h, and the diameters of the inhibition zones were measured.

Minimum inhibitory concentration (*MIC*) were determined to quantify antibacterial activity by using a broth micro dilution assay [38]. *MIC* values were measured in LB media using a 96-well plate. Serial dilution was performed, and the final concentrations of the film solutions were 50, 25, 12.5, 6.25, 3.75, and 1.5 μg/mL. Samples in LB media without a microbial suspension were considered as negative control samples. The 96-well plates were incubated at 37 °C for 24 h. The turbidity of the plate contents was interpreted as the growth of microorganisms and the appearance of the content of each well, as compared to the appearance of the negative control samples. The lowest concentration that exhibited no turbidity following incubation was interpreted as the *MIC*. To determine the minimum bactericidal concentration (*MBC*), the mixtures in each well that exhibited no turbidity were streaked onto MHA and incubated at 37 °C for 24 h. The lowest concentration of the test substance that prevented colony formation was considered as the *MBC*.

The optical densities of the bionanocomposite film solutions were analyzed at 600 nm for an antibacterial activity study. For each group, three pieces of 10 mm × 10 mm films were added to 1 mL of bacteria solution and incubated at 37 °C. The optical density at 600 nm (OD600) was measured every 3 h for 24 h in triplicate to analyze the growth of bacteria at different times.

Performance testing of the bionanocomposite films was conducted using a slightly modified process compared to the ASTM (ASTM E2149-0) and JIS methods (JIS Z 2801:2020). Each sample (50 mm × 50 mm) was sterilized under UV light, and 0.4 mL of each microbial suspension was dispersed onto the sample film and covered with a sterilized polyethylene film. Each sample was then incubated at 37 °C for 24 h. Next, the treated samples were carefully rinsed with a saline solution. For viable cell enumeration, a serial dilution of each microbial suspension was dispensed onto a tryptic soy agar plate, followed by incubation at 37 °C for 24 h. Antibacterial activity was calculated using the following equation:(3)$$Antibacterial\;activity\;\left(\%\right)=\frac{A-B}{A}\times 100$$
where *A* and *B* are the average numbers of viable cells in the control and test samples, respectively.

## 3. Results and Discussions

### 3.1. ZnO and Functionalized ZnO NP Characterization

The FT-IR spectra of as-prepared nanocomposites are presented in Figure 1. The FT-IR spectra of the pristine ZnO in Figure 1a exhibit absorption peaks in the ranges of 3300–3600 cm^−1^, 900–1200 cm^−1^, and 400–600 cm^−1^, which correspond to the stretching vibrations of OH, carboxyl/alkoxyl C-O, and Zn-O, respectively. The PANI-functionalized ZnO nanocomposites exhibit the same prominent peaks as pristine ZnO with two additional peaks at approximately 1632.2 cm^−1^ (Figure 1b) and 1384.2 cm^−1^ (Figure 1c), which correspond to N-H (primary amine) bending and C-N stretching modes, respectively. These results indicate that the PANI molecules were successfully functionalized on the ZnO surfaces by the impregnation process [37].

According to the XRD diffraction peaks in Figure 2, all of the samples exhibit a hexagonal crystal wurtzite structure (JCPDS card 89-13971). Characteristic peaks at 31.7°, 34.4°, 36.2°, 47.5°, 56.5°, 62.8°, 66.3°, 67.9°, 69.0°, 72.5°, and 76.9° correspond to the (100), (002), (101), (102), (110), (103), (200), (112), (201), (004), and (202) lattice planes, respectively. The crystallite sizes presented in Table 1 were calculated by using the Scherer’s equation to determine the average crystallite sizes (*D*) of the samples [39] as follows:(4)$$D=(\frac{0.9\lambda }{\beta cos\theta }),$$
where *λ* is the incident X-ray wavelength, *β* is the full width at half maximum intensity, and *θ* is the Bragg’s angle. There are slight changes in average crystal size and particle size following the PANI functionalization of ZnO. However, PANI functionalization does not affect the morphology or lattice constant of ZnO. The lattice parameters (*a* and *c*) were calculated using the following hexagonal formulation [40]:(5)$$\frac{1}{d_{hkl}^{2}}=\left[\frac{4}{3}\left(h^{2}+hk+k^{2}\right)+l^{2}{\left(\frac{a}{c}\right)}^{2}\right]\frac{1}{a^{2}},$$
where *d_hkl_* is the distance between adjacent planes with Miller–Bravais indices (*hkl*)

The surface morphologies of the PANI-functionalized ZnO nanocomposites were studied by using FE-SEM, as shown in Figure 3a–d. In Figure 3a, the pristine ZnO nanocomposites exhibit a uniform distribution of nanorods mixed with NPs. Compared to the pristine ZnO, the particle shapes of the as-prepared nanocomposites with PANI functionalization (Figure 3b–d) are similar with a slightly increased particle size, as indicated in Table 1. High-resolution transmission electron microscopy was utilized to analyze the crystal structures of the as-prepared nanocomposites, as shown in Figure 3e–h. The insets in Figure 3e–h reveal well-resolved lattice planes corresponding to the (100) planes of hexagonal wurtzite ZnO crystals.

The XPS spectra and binding energy values of the ZnO and functionalized ZnO samples are presented in Figure 3i. Figure 3i presents the XPS spectra of ZP0 and ZP3 with the appearance of C, O, N, and Zn constituents. Figure 3k–l presents the deconvoluted XPS spectra of the N 1s peak. The functionalized ZnO samples exhibit C-N and C-NH^3+^ peaks at 399.81 eV and 401.72 eV, respectively, which confirm the appearance of PANI of the ZnO surface [40]. Additionally, a shifting peak of Zn 2p can be observed following PANI functionalization, as shown in Figure 3j. This peak corresponds to Zn 2p_1/2_ from 1022.28 eV (ZP0) to 1022.48 eV (ZP3) and to Zn 2p_3/2_ from 1045.38 eV (ZP0) to 1045.58 eV (ZP3) [41,42,43]. These results support the occurrence of chemical interactions between the PANI molecules and ZnO surfaces. In combination with the FT-IR results, these results demonstrate the effectiveness of PANI functionalization on ZnO surfaces.

Table 1 presents the zeta potential distributions for all samples. It is well known that wurtzite-type ZnO NPs have positive surface charges in the as-prepared state [42]. Following PANI functionalization, the zeta potential values (*ζ*) shift from 17.50 (ZP0) mV to 25.87 mV (ZP3). Zeta potential measurements were conducted five times for each sample, and it was determined that surface charge modification using PANI mainly affects the electrostatic activity between positively charged PANI and ZnO surface charge.

### 3.2. Bionanocomposite Films Characterization

Figure 4 presents SEM micrographs and energy-dispersive X-ray spectroscopy (EDS) mappings of CS films and functionalized ZnO bionanocomposite films. The neat CS films provide an excellent polymerization ability based on its smooth surface and the absence of starch granules, indicating a well-designed film preparation process. However, all of the bionanocomposite films exhibit white aggregates and small holes dispersed over the film surfaces, which are attributed to the presence of ZnO crystals. All micrographs reveal homogeneous and relatively smooth film surfaces. Additionally, the presence of ZnO and N content from PANI is confirmed by the EDS mapping images. The intensity of N in the EDS micrograph of CS/ZP3 is greater than that in the CS/ZP2 and CS/ZP1 micrographs, which can be attributed to the greater concentration of PANI.

Figure 5a presents the XRD patterns of CS and bionanocomposite films. The pure CS film exhibits a typical A-type crystalline pattern with strong diffraction peaks at 2*θ* = 17.1° and 22.1°. These peaks are virtually invisible for the composite films. Additionally, one can see that the diffraction peaks of the composite films consisting of ZnO NPs and PANI-functionalized ZnO exhibit the same pattern as those of the pristine ZnO, which agrees with the results discussed in the previous section. Sharp peaks in the XRD patterns indicate high crystallinity without any impurities because all of the peaks are present in the XRD pattern matches (JCPDS card no. 89-13971). Furthermore, no diffraction peaks corresponding to PANI can be observed for any samples [44]. These results indicate that the intermolecular interactions between ZnO and CS during gelatinization via heating lead to a loss of crystallinity in the starch and that a starch solution consisting of acetic acid and glycerol provides excellent compatibility [45].

Figure 5b presents FT-IR spectra of samples that were captured to investigate interactions in the bionanocomposite films. The band from approximately 3300 to 3600 cm^−1^ corresponds to OH vibration groups on the surfaces of the ZnO NPs, and the peak between 400 and 600 cm^−1^ corresponds to the stretching vibrations of Zn-O. Additionally, the spectra of all samples contain peaks for the functional groups of CS in the ranges of 1100–1150 cm^−1^, 2340–2960 cm^−1^, and 2355–2155 cm^−1^, which correspond to C-O stretching, C-H stretching, and glycerol [31,46]. The stretching in the ranges of 1020–1030 cm^−1^ and 1370–1390 cm^−1^ is related to C-O-C and C=O functional groups, respectively [47].

### 3.3. Functional and Water Susceptibility of Bionanocomposite Films

The thickness of bionanocomposite films was measured in micrometers at five random points, and the average result was collected, as shown in Table 2. The CS film showed the least thickness (0.174 ± 0.001 mm). After the addition of ZnO and functionalized ZnO NPs, the bionanocomposite film thickness increased, due to the increase in the crystallite size of the ZnO NPs and functionalized ZnO NPs. This result could also attribute to the inner molecular interaction between starch and ZnO NPs [48].

Moisture content (*MC*), water solubility (*WS*), and water vapor permeability (*WVP*) of CS, CS/ZP0, CS/ZP1, CS/ZP2, and CS/ZP3 are presented in Table 2. The water susceptibility of the as-prepared bionanocomposite films is lower than that of the neat CS films. It was observed that the addition of pristine ZnO and functionalized ZnO significantly decreases the *MC* and *WS* values. These decreases are attributed to the interaction between the starch matrix and the ZnO NPs, which causes the higher binding with free water molecules to interact with the bionanocomposite films [49,50]. The *WVP* value of ZnO-NP-incorporated bionanocomposite films shows a lower values compared with neat CS films (5.32 ± 0.15 g·m/m^2^·s·Pa). In addition, the *WVP* values of the CS/ZP1, CS/ZP2, and CS/ZP3 (4.52 ± 0.33, 4.51 ± 0.35, and 4.49 ± 0.23 g·m/m^2^·s·Pa, respectively) films did not show a significant difference when compared with CS/ZP0 (4.59 ± 0.52 g·m/m^2^·s·Pa), respectively. The decrease in the *WVP* value of bionanocomposite films could be a consequence of the decreasing limit in the water vapor transmittance through the starch matrix, which could cause more interaction between water molecules in the starch matrix and hydrogen bond of ZnO and functionalized ZnO NPs, in agreement with the FT-IR results [8].

### 3.4. Mechanical Properties of Bionanocomposite Films

The tensile strength and percent elongation at break of the bionanocomposite films are presented in Table 3. The CS/ZP0 (7.51 ± 0.29 MPa) showed a significant increase in tensile strength, unlike the neat CS films (4.71 ± 0.27 MPa), owing to the uniform dispersion of ZnO NPs in the starch polymer matrix that causes a strong interaction between starch and ZnO NPs [51]. In the case of CS/ZP1 (6.07 ± 0.13 MPa), CS/ZP2 (6.11 ± 0.15 MPa), and CS/ZP3 (6.14 ± 0.09), the tensile strength was lower than that of CS/ZP0 but higher than that of the neat CS films. This is because the ZnO NPs after PANI functionalization have a higher density, which could be related to the decreasing tensile strength [51]. In addition, the elongation at break of the bionanocomposite films dramatically decreased from 96.79% to 69.86% with the addition of ZnO and functionalized ZnO NPs. However, the absence of a statistical difference in the case of an increasing concentration of functionalized ZnO NPs could be an effect of the restriction in the motion of the polymer and PANI chain in the bionanocomposite film matrix [52].

### 3.5. Thermal Properties of Bionanocomposite Films

Thermal degradation of bionanocomposite films was analyzed by using TGA, and the result is shown in Figure 6, which exhibits the TG curve. The weight loss could generally be divided into three regions. The initial weight loss below 100 °C corresponds to the evaporation of absorbed water. The second stage, following mass loss until the onset temperature of the thermal decomposition at around 300 °C, associated with water in bionanocomposite films. The final stage correlated with the amylose amylopectin chain degradation at around 350 °C [49,52]. The results indicate that, with the addition of pristine ZnO and functionalized ZnO NPs, the thermal stability of the sample increased while the weight loss decreased. Higher thermal stability can be attributed to the interaction with the nanocomposite surface and polymer matrix, in agreement with the crystallinity in the XRD result and other work [49]. In addition, the residual content in the bionanocomposite films after thermal decomposition was higher than that of the neat CS films [53].

### 3.6. Antibacterial Activity

The inhibitory zone diameters of the bionanocomposite films for both Gram-negative bacteria (*E. coli*) and Gram-positive bacteria (*S. aureus*) were determined using the disc diffusion method. As shown in Table 3 and Figure 7a,b, the neat CS film does not provide any antibacterial activity. However, the ZnO and functionalized ZnO bionanocomposite films provide inhibitory zones. The diameters of these zones are summarized in Table 3. CS/ZP3 provides the largest inhibitory zone diameters for *E. coli* (12.40 ± 0.46 mm) and *S. aureus* (11.47 ± 0.32 mm). For the *MICs* and *MBCs* of the film solutions, only the CS/ZP3 film was selected for additional study based on its large inhibitory zone diameters (>10 mm) [38]. The CS/ZP3 film solution exhibits the lowest *MICs* for *E. coli* at 12.5 μg/mL and *S. aureus* at 6.25 μg/mL, respectively. It also exhibits the lowest *MBCs*. As a result, it can be concluded that differences in the structures and components of the cell membranes of Gram-negative and Gram-positive bacteria can pose major challenges in terms of cell growth inhibition and antibacterial activity [54]. The growth curve in Figure 7c,d represents intermittent measurements of OD600. To demonstrate the potential antibacterial activity of the functionalized ZnO bionanocomposite films, an untreated microbial activity diagram is also included. One can see that all samples provide excellent antibacterial activity and efficiently inhibit the growth of bacterial cells. A lag phase for both bacterial growth curves could be observed for up to 6 h for the bionanocomposite films, which is longer than those for the control sample, indicating that the bionanocomposite films significantly inhibit bacterial cell multiplication. The lag phase of *E. coli* is shorter than that of *S. aureus*. In Figure 7c,d, one can see that the stationary phase of *E. coli* begins at 12 h, but the stationary phase of *S. aureus* begins at approximately 15 h. For the same sample conditions and timeframe, the bionanocomposite films exhibit excellent antibacterial activity for *E. coli* over *S. aureus*. The antimicrobial efficiency of the functionalized ZnO nanocomposite films are slightly different from those measured using the ASTM (ASTM E2149-0) and JIS methods (JIS Z 2801:2020). We tested these methods with both bacterial suspensions, and the results are summarized in Table 3. A pure CS film was prepared as a control sample. As the functionalized ZnO content increases, the antimicrobial activity of the bionanocomposite films increases, with CS/ZP3 exhibiting the highest values of 76.15% and 72.36% for *E. coli* and *S. aureus*, respectively

Based on these results, we can describe the antibacterial activity mechanism of ZnO NPs by referencing previous studies [4,23]. The antibacterial activity of metal NPs stems from various factors, including the generation of ROS, the release of cationic ions, and cell wall damage. In the case of ZnO NPs, many researchers have reported that Zn^2+^ ions can pass through the cell walls of bacteria and interact with cytoplasmic content to kill the bacteria [55,56,57]. We aimed to elucidate the interactions at the interface between functionalized ZnO NPs and bacteria. It is well known that bacteria cell walls have negative surface charges. In the case of Gram-positive bacteria, this negative charge stems from teichoic acid linked to a thick peptidoglycan membrane, where this organic acid and membrane have negative surface charge potentials [58,59]. However, in Gram-negative bacteria, the thin peptidoglycan membrane is covered by an outer membrane of phospholipid and lipopolysaccharide, which generates a negative surface charge of [59,60,61]. Therefore, functionalizing ZnO NPs with PANI increases their positive charges, resulting in enhanced antibacterial activity. According to the positive charge of functionalized ZnO and negative charges of bacteria cell walls, our experimental results based on CS films revealed that the antibacterial activity of bionanocomposite films is stronger for Gram-negative bacteria than for Gram-positive bacteria. This result agrees with other reports that have suggested that ZnO NPs have the potential to decrease *E. coli* growth rates via the agitation of *E. coli* cell membranes [62]. From our perspective, the antibacterial activity of PANI-functionalized ZnO NPs could stem from both the production of ROS and the consolidation or substitution of electrostatic interactions between ZnO NPs and bacterial cell membranes.

## 4. Conclusions

A PANI-functionalized ZnO surface was prepared using a simple and green impregnation technique. The PANI content increased the positive surface charges of the ZnO NPs, which were applied to a biopolymer film to improve antibacterial activity. The antibacterial activity of the CS/ZP3-functionalized ZnO bionanocomposite film was excellent for both Gram-negative (*E. coli*) and Gram-positive bacteria (*S. aureus*), which can be attributed to the strong positive charges of the bionanocomposite film interacting with the negatively charged bacterial cell walls. Based on the data presented in this study, these films would be an easy accessible and low-cost surface charge modification approach for the development of single use antibacterial-biodegradable thermoplastic packaging.

## Figures and Tables

**Figure 1 polymers-13-00425-f001:**
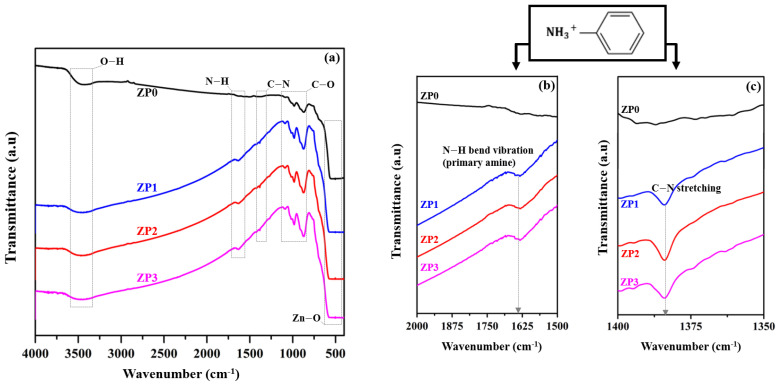
FT-IR of zinc oxide (ZnO) and polyaniline (PANI)-functionalized ZnO samples: (**a**) 400–4000 cm^−1^, (**b**) N-H bending mode region: 1500–2000 cm^−1^, and (**c**) C-N stretching mode region: 1350–1400 cm^−1^.

**Figure 2 polymers-13-00425-f002:**
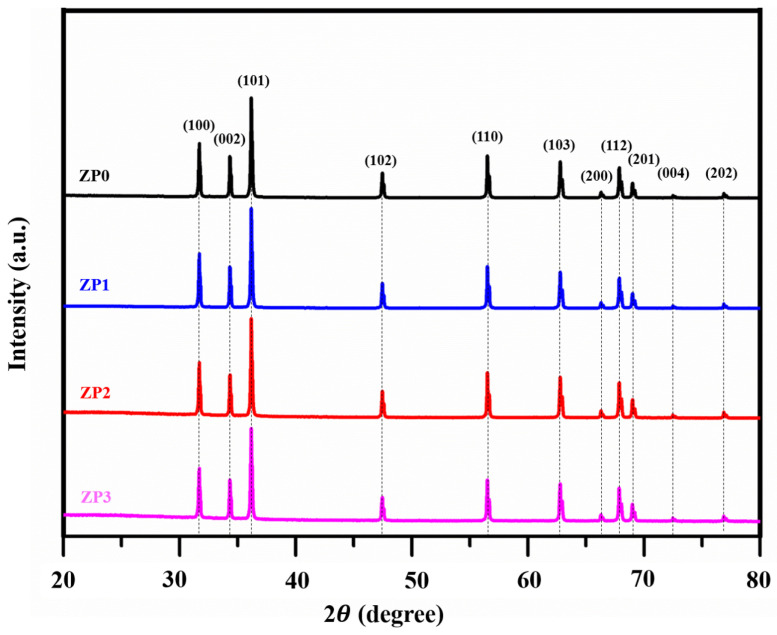
XRD patterns of ZnO and functionalized ZnO samples.

**Figure 3 polymers-13-00425-f003:**
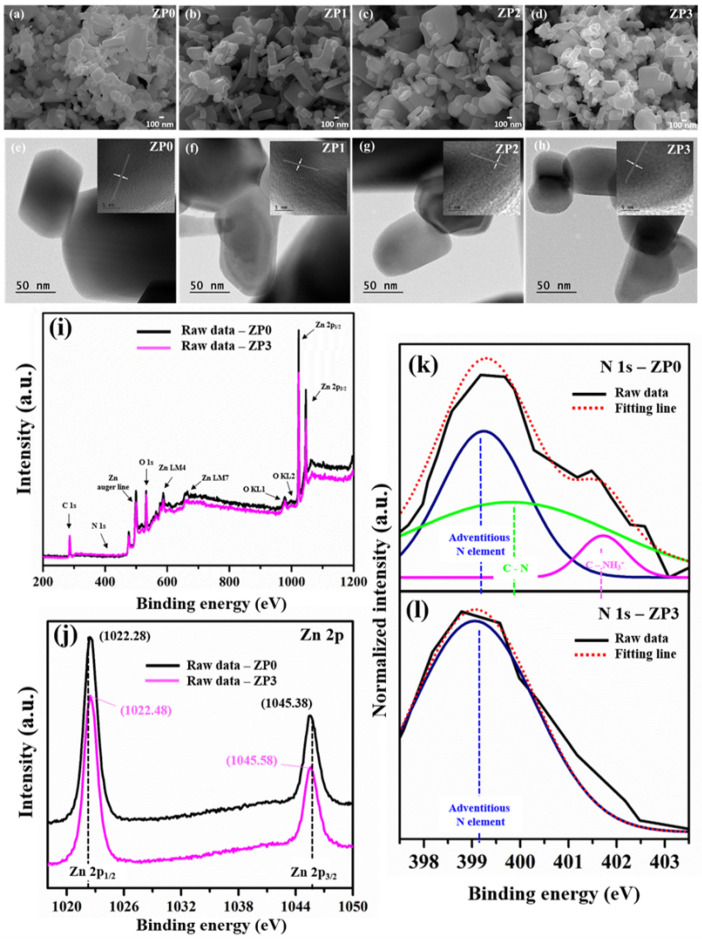
(**a**–**d**) FE-SEM and (**e**–**h**) HR-TEM images of ZnO and functionalized ZnO samples and XPS of ZP0 and ZP3 samples; (**i**) XPS Survey spectrum, (**j**) XPS Zn 2p and, (**k**) and (**l**) deconvoluted XPS N 1s of ZP0 and ZP3, respectively.

**Figure 4 polymers-13-00425-f004:**
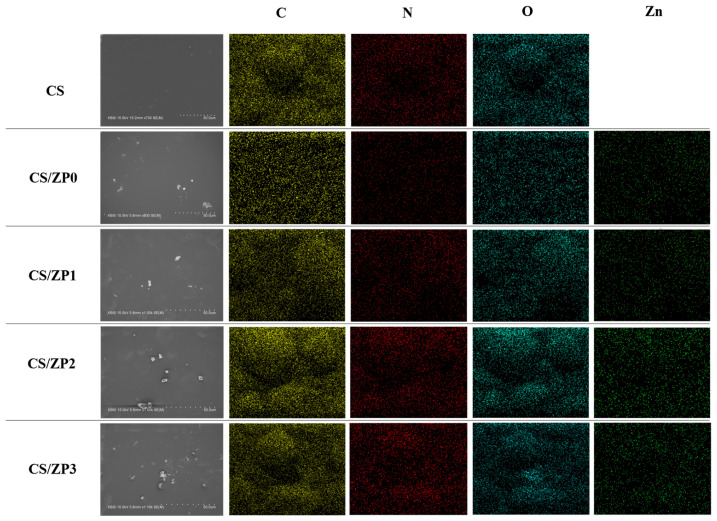
FE-SEM with EDS mapping image of neat corn starch (CS) film and bionanocomposite film samples.

**Figure 5 polymers-13-00425-f005:**
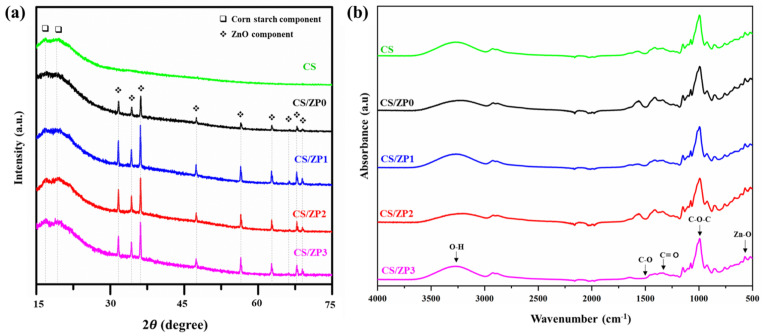
(**a**) XRD patterns of neat CS film and bionanocomposite film samples and (**b**) FT-IR spectra of neat CS film and bionanocomposite film samples.

**Figure 6 polymers-13-00425-f006:**
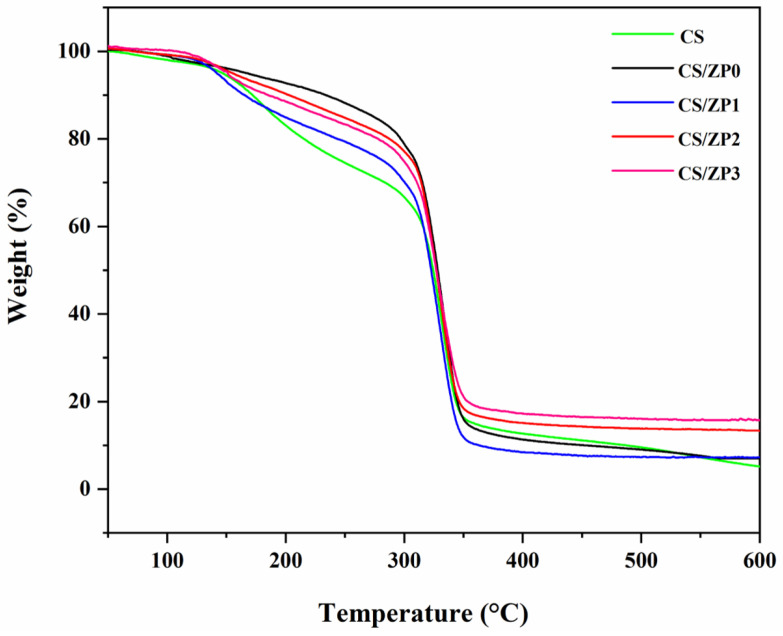
Thermogravimetric analysis of the neat CS film and bionanocomposite film samples.

**Figure 7 polymers-13-00425-f007:**
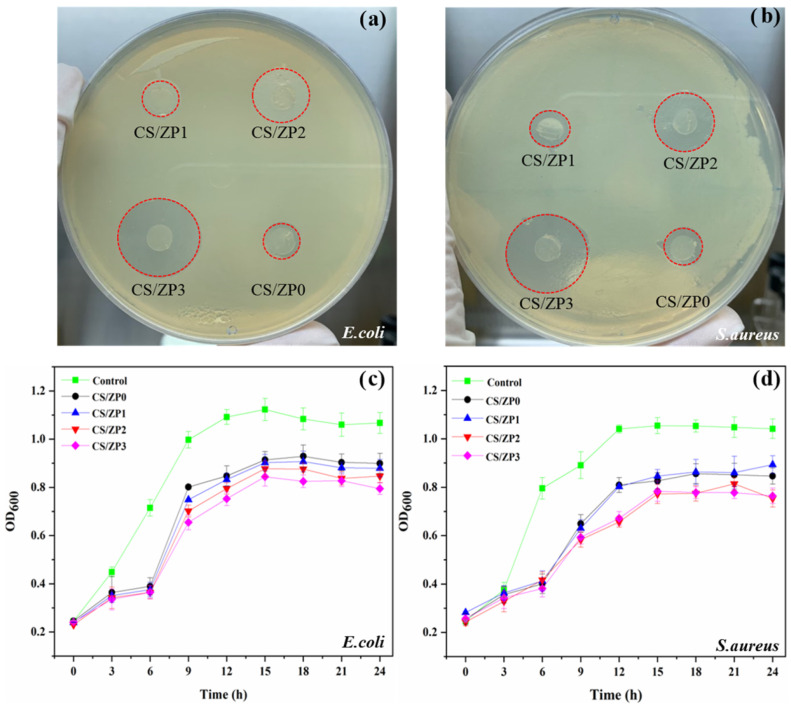
Inhibitory zone of bionanocomposite films against (**a**) *E. coli* and (**b**) *S. aureus* and optical density measurement of bacteria at a wavelength of 600 nm (OD600) for (**c**) *E. coli* and (**d**) *S. aureus*.

**Table 1 polymers-13-00425-t001:** Summaries of crystal information and surface charge of ZnO and functionalized ZnO samples.

Sample	XRD			Zetasizer Nano ZS	
	Lattice Constant (Å)		Crystal Size (nm)	Average Size (nm)	Zeta Potential (mV)
	a	c			
ZP0	3.253	5.213	95.08 ± 5.52 *^a^*	198.23 ± 7.22 *^c^*	17.50± 0.46 *^d^*
ZP1	3.253	5.213	84.00 ± 6.90 *^c^*	236.95 ± 6.86 *^b^*	22.63 ± 0.31 *^c^*
ZP2	3.253	5.213	92.96 ± 5.76 *^ab^*	262.69 ± 5.54 *^a^*	24.77 ± 0.31 *^b^*
ZP3	3.253	5.213	89.85 ± 8.06 *^bc^*	246.20 ± 5.69 *^a^*	25.87 ± 0.55 *^a^*

Note: Values in the same column followed by different letters are significantly different (*p* < 0.05) according to Duncan’s multiple range tests.

**Table 2 polymers-13-00425-t002:** Functional properties of bionanocomposite films.

Sample	Thickness (mm)	Moisture Content (%)	Water Solubility (%)	Water Vapor Permeability (g·m/m^2^·s·Pa)
CS	0.174± 0.001 *^c^*	19.47 ± 0.32 *^a^*	34.76 ± 0.78 *^a^*	5.32 ± 0.15 *^a^*
CS/ZP0	0.182 ± 0.002 *^b^*	16.68 ± 0.25 *^d^*	27.68 ± 0.81 *^b^*	4.59 ± 0.52 *^b^*
CS/ZP1	0.186 ± 0.003 *^ab^*	18.41 ± 0.38 *^b^*	26.58 ± 0.75 *^c^*	4.52 ± 0.33 *^b^*
CS/ZP2	0.186 ± 0.004 *^ab^*	18.17 ± 0.35 *^bc^*	26.30 ± 0.56 *^c^*	4.51 ± 0.35 *^b^*
CS/ZP3	0.188 ± 0.004 *^a^*	17.89 ± 0.41 *^c^*	26.14 ± 0.15 *^c^*	4.49 ± 0.23 *^b^*

Note: Values in the same column followed by different letters are significantly different (*p* < 0.05) according to Duncan’s multiple range tests.

**Table 3 polymers-13-00425-t003:** Mechanical properties, diameters of inhibition zones, and antibacterial activity of CS film and bionanocomposite film samples against *E. coli* and *S. aureus*.

Sample	Mechanical Properties		Inhibitory Diameter (mm)		% Reduction	
	Tensile Strength (MPa)	Elongation %	*E. coli*	*S. aureus*	*E. coli*	*S. aureus*
CS	4.71 ± 0.27 *^c^*	96.79 ± 2.11 *^a^*	0.00 ± 0.00 *^d^*	0.00 ± 0.00 *^d^*	0.00	0.00
CS/ZP0	7.51 ± 0.29 *^a^*	75.19 ± 2.15 *^b^*	8.90 ± 0.10 *^c^*	8.53 ± 0.21 *^c^*	65.76	62.71
CS/ZP1	6.07 ± 0.13 *^b^*	71.02 ± 1.99 *^c^*	8.93 ± 0.38 *^c^*	8.90 ± 0.20 *^c^*	68.14	67.35
CS/ZP2	6.11 ± 0.15 *^b^*	70.07 ± 1.38 *^c^*	9.87 ± 0.35 *^b^*	9.80 ± 0.40 *^b^*	71.04	68.09
CS/ZP3	6.14 ± 0.09 *^b^*	69.86 ± 1.20 *^c^*	12.40 ± 0.46 *^a^*	11.47 ± 0.32 *^a^*	76.15	72.36

Note: Values in the same column followed by different letters are significantly different (*p* < 0.05) according to Duncan’s multiple range tests.

## Data Availability

Not applicable.

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
