# Peer review of "Investigation of Functionalized Surface Charges of Thermoplastic Starch/Zinc Oxide Nanocomposite Films Using Polyaniline: The Potential of Improved Antibacterial Properties"

_polymers, 2021, doi:10.3390/polym13030425_

Round 1

Reviewer 1 Report

The article is well written, easy to read, the research has novelty. There is some minor revision necessary, regarding typing and editing mistakes. For example, when you described the materials used, the producers should be written  using capital letters (for the first letter of the word).

Author Response

Revised manucript is attached in PDF file

Reviewer 2 Report

Manuscript presents several fundamental flaws.

  1. FTIR results revealed serious concerns: spectra does not present proper resolution; peak assigned to C-O vibration cannot be identified; spectra only suggest the presence of PANI and cannot be used to confirm the proposed functionalization, as suggested by authors (lines 191 - 192); FTIR spectra usually starts with high wavenumbers in the x-axis.
  2. XRD results do not provide significant data;
  3. XPS resuls:

a. Line 219. The functionalized ZnO samples exhibit C-N and C-NH3+ 220 peaks at 399.81 eV and 401.72 eV, respectively, which confirm the appearance of PANI of the ZnO surface [41].

Reference list does not contain the reference 41.

b. Line 221. Additionally, a shifting peak of Zn 2p can be observed following PANI functionalization, as shown in Figure 3B (b). This peak corresponds to Zn 2p1/2 from 1022.28 eV (ZP0) to 1022.48 eV (ZP3) and to Zn 2p3/2 from 1045.38 eV (ZP0) to 1045.58 eV (ZP3) [42]. These results support the occurrence of chemical interactions between the PANI molecules and ZnO surfaces. 

Reference 42 does not provide evidences that both peak shiftings could be used to confirm the functionalization. This reference was not properly cited.

4. Line 34-36. The incorporation of ZnO NPs has been reported to have a positive effect on the antibacterial activity of biopolymers, including soy protein isolated, chitosan, and starch [4,7,19,20].

Since the effects of ZnO NP incoporation on antibacterial activity of starch has been already investigated, this manuscript does not provide any novelty. Despite the several fundamental flaws.

Author Response

(The authors gave the same response as above.)

Round 2

Reviewer 2 Report

Accept in the present form.

 form.